# Assessing the Efficacy of PLGA-Loaded Antimicrobial Peptide OH-CATH30 Microspheres for the Treatment of Bacterial Keratitis: A Promising Approach

**DOI:** 10.3390/biom13081244

**Published:** 2023-08-12

**Authors:** Xiaoqian Jiao, Xufeng Dong, Hu Shan, Zhihua Qin

**Affiliations:** College of Veterinary Medicine, Qingdao Agricultural University, Qingdao 266109, China; xqj426@stu.qau.edu.cn (X.J.); dongxufeng@qau.edu.cn (X.D.); hush@qau.edu.cn (H.S.)

**Keywords:** OH-CATH30, PLGA, microspheres, tolerance, keratoconus, treatment

## Abstract

Bacterial keratitis in animals presents challenges due to ocular structural barriers, hindering effective drug delivery. In this study, we used biocompatible and biodegradable poly(lactic-co-glycolic acid) (PLGA) to encapsulate the naturally occurring antimicrobial peptide OH-CATH30, an alternative to conventional antibiotics, for the treatment of bacterial keratitis in animals. Microspheres (MS) were prepared using a modified water-in-oil-in-water (W/O/W) double-emulsion method with optimized osmotic pressure. We conducted comprehensive evaluations, including in vitro characterization, encapsulation efficiency determination, in vitro release kinetics, and in vivo/vitro assessments of irritation and bacterial inhibition. The optimized method yielded microspheres with impressive encapsulation efficiency of 75.2 ± 3.62% and a loading capacity of 18.25 ± 5.73%, exhibiting a well-defined particle size distribution (200–1000 nm) and a ζ-potential of −17.3 ± 1.91 mV. The microspheres demonstrated initial burst release followed by sustained and controlled release in vitro. Both in vitro and in vivo tolerance tests confirmed the biocompatibility of the drug-loaded microspheres, as they did not elicit significant irritation in ocular tissues. Remarkable antibacterial effects were observed in both in vitro and in vivo experiments. Our developed PLGA microspheres show promise as an alternative therapeutic option for topical administration in managing keratitis, offering exceptional drug delivery capabilities, improved bioavailability, and potent antibacterial efficacy.

## 1. Introduction

Bacterial keratitis (BK) is a common disease in veterinary ophthalmology and one of the leading causes of blindness in dogs and cats [1]. The rapid progression of the disease can result in devastating corneal infections, leading to severe visual impairment [2,3]. However, the administration of ocular medication in pets demonstrates significant homogeneity, presenting ongoing challenges for conventional pharmacologic intervention strategies in their quest to surmount microbial resistance to ocular antibiotics [4]. Therefore, in the current scenario, it is imperative to search for effective alternative methods to kill intracellular bacteria [5].

Antimicrobial peptides are widely found in nature and serve as essential components of the innate immune system in most organisms [6]. Due to their broad antimicrobial activity and low resistance, antimicrobial peptides have emerged as excellent alternatives to traditional antimicrobial drugs [7]. OH-CATH30, a cathelicidin antimicrobial peptide derived from the king cobra, exhibits remarkable antimicrobial, anti-inflammatory, and immunomodulatory properties [8,9]. Notably, OH-CATH30 demonstrates broad-spectrum antimicrobial activity against a diverse range of bacteria, comparable to amikacin for Gram-positive bacteria and surpassing polymyxin B against nearly all tested Gram-negative bacteria [10]. Previous studies by Li et al. explored the potential of topical OH-CATH30 application as a promising therapy for Pseudomonas aeruginosa ocular bacterial infections [11]. However, the widespread use of natural peptides is hindered by their susceptibility to degradation and short half-life. Moreover, the unique structure of the eye poses challenges in achieving optimal drug bioavailability [12].

To address these limitations, an innovative approach has been proposed, utilizing drug-carrier-encapsulated antimicrobial peptide technology for the controlled release and protection of antimicrobial peptides from enzymatic digestion [13]. Among these polymers, poly(lactic-co-glycolic acid) (PLGA) delivery systems are commonly employed for in vivo drug delivery due to their extensive safety record [14,15]. Choi et al. conducted a study demonstrating that injectable PLGA microspheres, incorporating the WKYMVM peptide, exhibited sustained release, promoting blood flow restoration and capillary growth in a hind limb ischemia model [16]. Additionally, Rosas et al. showed that a PLGA microsphere-based delivery system enhanced the immunogenicity of the SPf66 vaccine without compromising its integrity and immunogenicity [17]. These studies highlight the promising capabilities of polymer-based drug delivery systems utilizing PLGA as a versatile platform for drug loading. In the context of preparing PLGA microspheres, the double-emulsion method (W/O/W) is commonly used and offers advantages such as a uniform particle size distribution, improved colloidal stability, reduced production times and costs, and increased yields [18,19]. Depending on the preparation process, the structures of microspheres may vary, with drugs either encapsulated in the matrix or adsorbed on the surface [20,21].

Material-based drug delivery systems have been widely utilized to enhance the bioavailability of bioactive proteins and peptides for the treatment of ocular diseases [22]. However, limited research has been conducted on the topical application of natural antimicrobial peptides to microsphere-based drug delivery systems for bacterial keratitis treatment [23,24]. In this study, our main objective was to achieve the longer sustained release of OH-CATH30 by encapsulating it in poly(lactic-co-glycolic acid) (PLGA), thus addressing the short half-life, easy inactivation, and degradation issues associated with OH-CATH30. We thoroughly evaluated the in vitro release behavior, as well as the physical, ocular tolerance (in vivo and in vitro), and antimicrobial properties of PLGA-encapsulated OH-CATH30. Based on their favorable physical and bactericidal properties, PLGA-encapsulated OH-CATH30 microspheres (PLGA-OH-CATH30 MS) show great potential as a promising alternative for the effective treatment of ulcerative bacterial keratitis in animals.

## 2. Materials and Methods

### 2.1. Materials

Biodegradable poly(d, l-lactic acid) ((50:50), MW 40000, acid-terminated) was purchased from Chongqing Yu-si Pharmaceutical Technology Co., Ltd., Chongqing, China. Poly(vinyl alcohol-1788), with alcohol solubility of 87.0~89.0% (mol/mL), was purchased from Shanghai Macklin Biochemical Co., Ltd., Shanghai, China. Dichloromethane (HPLC-grade), acetic acid (HPLC-grade), acetonitrile (HPLC-grade), and NaCl (GR, 99.8%) were purchased from Sinopharm Pharmaceutical Co., Beijing, China. Peptide OH-CATH30 (KFFKKLKNSVKKRAKKFFKKPRVIGVSIPF) was synthesized by the solid-phase method by Anhui Guotai Biotechnology Co., Ltd., Hefei, China.

### 2.2. Identification of Antimicrobial Peptides

The purity of the peptides was assessed using HPLC (the chromatographic column was the YMC-Triart C18 (4.6 mm × 250 mm, 5 μm).. Pump A contained 0.1% trifluoroacetic acid in 100% water, and pump B contained 0.1% trifluoroacetic acid in 100% acetonitrile. The elution conditions were set as A:B = 20:80 (*v*/*v*) with a flow rate of 1 mL/min. The wavelength was set at 214 nm, and the injection volume was 25 μL. The column temperature was maintained at 30 °C. The peptide products were subjected to analysis through matrix-assisted laser desorption ionization time-of-flight mass spectrometry.

### 2.3. OH-CATH30-PLGA Preparation of MS

PLGA-OH-CATH30 MS were prepared using the water-in-oil-in-water (W/O/W) double-emulsion method, with optimization of the preparation process to create three different types of internal aqueous-phase osmotic-pressure MS. In brief, 30 mg of PLGA was dissolved in 5 mL of dichloromethane, and 10 mg of OH-CATH30 (molecular weight: 3593.80 g/mol) was dissolved in 1 mL of sterile, enzyme-free water. The aqueous-phase peptide solution was added dropwise to the organic polymer phase, forming the primary emulsion (W1/O), intermittently sonicated for 60 s under an ice water bath at an energy output of 100 W amplitude. The primary emulsion was then added dropwise with a syringe to 10 mL of 0.75% (*w*/*v*) PVA solution containing varying concentrations of NaCl (0.1%, 0.6%, 0.9%, and 1.2%). The mixture was sonicated for 30 s under an ice water bath at an energy output of 60 W amplitude to create a W/O/W solution. Subsequently, the W/O/W solution was added to 15 mL of 0.5% PVA solution and stirred at 400 rpm for 4 h at room temperature. This allowed the MS to solidify and the organic solvent to evaporate. The MS were collected in an aqueous medium through repeated low-speed centrifugation (3000 rpm, 15 min, 4 °C), washed to remove excess PVA, and then lyophilized for 48 h. They were subsequently stored at −20 °C until further use.

### 2.4. Encapsulation Efficiency (EE) and Loading Capacity (LC)

The EE and LC of PLGA-OH-CATH30 MS were assessed using a centrifugation–HPLC procedure [25]. Briefly, the prepared MS were washed and subsequently lyophilized. The lyophilized MS were weighed to determine their dry weight. Meanwhile, the supernatant obtained from the centrifugal wash was collected and brought to a final volume of 50 mL. The amount of unbound peptides in the supernatant was quantified using HPLC, as described in Section 2.2.

The encapsulation efficiency Equation (1) and loading capacity Equation (2) were, respectively, calculated as follows:EE (%) = (total amount of peptides − amount of unbound peptides)/(total amount of peptides) × 100%;(1)
LC (%) = (total amount of drug − amount of unbound drug)/(MS weight) × 100%.(2)

### 2.5. MS Characterization Analysis

The surface morphology of the PLGA-OH-CATH30 MS was analyzed using a Sharp Corporation JSM-7500F scanning electron microscope (SEM) after coating them with a conductive resin using an E-1010 ion sputterer.

The lyophilized MS were dispersed in distilled water and sonicated to ensure uniform dispersion. Dynamic light scattering (DLS) measurements were performed using a Malvern NANO ZS900 instrument to evaluate the average particle size and polydispersity index (PDI) of the MS. Electrophoretic light scattering (ELS) measurements were also performed by this instrument to evaluate the zeta potential of the MS. Each batch was tested in triplicate.

### 2.6. MS In Vitro Release Experiment

The experimental procedure was extensively described in previous works [26]. Briefly, 10 mg of lyophilized PLGA-OH-CATH30 MS were suspended in 10 mL of phosphate-buffered saline (PBS; pH = 7.4) within centrifuge tubes. The tubes were incubated in a shaking incubator at 120 rpm and 37 °C. At predetermined time intervals (0.5 h, 2 h, 4 h, 8 h, 12 h, 24 h, 48 h, 3 d, 6 d, 9 d, 12 d, 15 d, 18 d, 21 d, 24 d, 27 d, and 30 d), 1 mL of the sample was extracted. Subsequently, the samples were centrifuged at 10,000 rpm for 10 min at 4 °C to separate the supernatants. The collected supernatants were analyzed by HPLC following the method described in Section 2.2. To maintain a constant volume, the supernatants were replenished with an equal volume of fresh and prewarmed PBS (pH = 7.4, 37 °C) after sampling. The in vitro release experiment was performed in triplicate to ensure the accuracy and reproducibility of the results.

The release rate was calculated according to Equation (3):Release rate (%) = (amount of drug released/total drug amount) × 100%.(3)

### 2.7. Bactericidal Activity

The Oxford cup diffusion method was utilized to evaluate the comparative inhibitory effects of the encapsulated antibacterial peptides, following the procedure described by [27]. The bacterial broth containing three activated strains (*E. coli* ATCC 8099, *S. aureus* ATCC 25923, and *P. aeruginosa* ATCC 27853) was centrifuged at 4000 rpm for 10 min. After centrifugation, the supernatant was decanted, and the pellet was resuspended in saline to achieve turbidity of 0.5 McFarland standard (approximately 1 × 10^8^ CFU/mL). Sterile nutrient agar medium was then carefully poured into Petri dishes at approximately 50 °C to ensure an even distribution of approximately 15 mL per dish. Once the agar solidified, 100 μL of the activated dilutions of the three bacterial strains were evenly spread on the agar surface and allowed to air-dry. Three sterilized forceps were used to position the Oxford cups evenly on each nutrient agar plate, ensuring proper placement. To investigate the inhibitory potential, separate sterilized Oxford cups were loaded with 100 μL of blank PLGA MS, 100 μL of OH-CATH30 (1 mg/mL), and 100 μL of PLGA-OH-CATH30 MS. The plates were then incubated in a constant-temperature incubator set at 37 °C for 24 h to monitor and document the observed inhibitory effect.

A modified time-kill assay, based on a previously reported method [28], was conducted to evaluate the antimicrobial activity as follows. The bacterial suspension was diluted to a concentration of 1 × 10^3^ CFU/mL using sterile saline. Subsequently, the PLGA-OH-CATH30 MS, prepared at a concentration of 1 mg/mL, were added to the bacterial solution and incubated for 4, 8, 12, and 24 h. Following the respective incubation periods, 20 μL of the mixture was aseptically inoculated onto a solid agar medium and evenly spread using disposable sterile applicator sticks. The Petri dishes were then inverted and placed in a 37 °C incubator after a 0.5 h incubation period. The number of bacterial colonies on each plate was meticulously observed. As a control group, only the bacterial suspension without the addition of PLGA-OH-CATH30 MS was included. Each experiment was conducted in triplicate, and the inhibition rate [29] was calculated accordingly.

The inhibition rate was calculated according to Equation (4):Inhibition rate (%) = [(number of bacterial colonies in the control group − number of bacterial colonies in the sample group)/number of bacterial colonies in the control group] × 100%.(4)

### 2.8. Chorioallantoic Membrane–Trypan Blue Staining Experiment (CAM-TBS)

The ocular irritation potential of the test samples was assessed using the chorioallantoic membrane test with trypan blue staining (CAM-TBS) method, as previously described [30]. Fertilized chick embryos were cultured until the tenth day, and a 2 cm × 2 cm opening was carefully created above the air chamber. Subsequently, the overlying membrane was removed with forceps, exposing the chorioallantoic membrane after infiltration with saline drops. A 300 µL drop (OH-Cath30 and PLGA-OH-CATH30 MS) of the test article was applied to the chorioallantoic membrane, allowing for a 5 min contact period. After the designated contact time, the membrane was rinsed thoroughly with saline solution to remove any residual test article. The stained chorioallantoic membrane was then excised and placed in 1 mL of formamide for 24 h to extract the absorbed material. The absorbance of the extract was measured at 611 nm, and the concentration of absorbed trypan blue was determined. The experimental setup included negative controls (saline) and positive controls (0.1 mol·L^−1^ NaOH solution). The degree of irritation induced by the test samples was quantitatively evaluated based on the absorbance of trypan blue, with higher absorbance values indicating greater ocular irritation potential.

### 2.9. Animal Experiments

#### 2.9.1. Animal Sources

The experimental rabbits weighing 2.0 and 3.0 kg were sourced from the Experimental Animal Center of Qingdao Agricultural University. Before inclusion in the study, the rabbits underwent a thorough screening process to ensure the absence of ocular inflammation or any visible abnormalities. The animals were individually housed in a controlled environment with a regulated temperature and lighting and access to water and food. Throughout the study, the welfare and treatment of the rabbits adhered to the guidelines outlined in the Association for Research in Vision and Ophthalmology (ARVO) resolution, which governs the ethical use of animals in ophthalmic research.

#### 2.9.2. Ocular Irritation Study

A total of 9 rabbits were randomized into 3 groups of 3 rabbits each. The left eye of each rabbit was treated with 50 μL of physiological saline as a control. In the right eye, one group of rabbits received 50 μL of OH-CATH30 (1 mg/mL) solution, another group received 50 μL of PLGA-OH-CATH30 MS (1 mg/mL) solution, and the last group received 50 μL of commercially available pet-specific Tarivid ophthalmic solution (containing 1 mg/mL of Ofloxacin). Each rabbit received ocular administrations three times a day for seven consecutive days. Throughout the study period, the rabbits’ eyes were systematically examined at specific time points to evaluate conjunctival redness, discharge, and signs of inflammation. Each eye was scored by three examiners who were blind to the treatment groups, and the resulting scores were averaged to quantify the degree of eye irritation using the ‘Draize Eye Irritation Test Scale’ [31].

#### 2.9.3. In Vivo Treatment

A strain of *S. aureus* (ATCC 25923) was selected to investigate the in vivo antimicrobial effects of antimicrobial peptides on bacterial keratitis in rabbit experiments. Bacterial keratitis was induced in both eyes of the rabbits using a carefully conducted procedure. The rabbits were first administered general anesthesia with 5 μg/kg dexmedetomidine, followed by local anesthesia with 0.2% lidocaine hydrochloride. A precise volume of 10 μL saline resuspension containing 1 × 10^3^ CFU/mL of *S. aureus* was then injected into the corneal stroma using a 30G needle, effectively inducing corneal infection. The experiment involved 15 rabbits, randomly distributed into five treatment groups (Groups I–V) by an impartial researcher, who was not involved in the examination and scoring procedures. Each group consisted of three animals, totaling six eyes. Groups I, II, and III received OH-CATH30 solution (1 mg/mL), PLGA-OH-CATH30 MS suspension (1 mg/mL), and commercially available pet-specific Tarivid ophthalmic solution (containing 1 mg/mL of Ofloxacin), respectively. The IV group served as the control and received saline treatment, while the V group remained uninfected and untreated. Eye drop administrations were performed three times daily for five consecutive days.

#### 2.9.4. Evaluation of Treatment

During the five-day treatment period, daily assessments were conducted to evaluate the efficacy of different treatments for bacterial keratitis. The rabbits’ eyes underwent meticulous examination using a slit lamp to identify key clinical indicators of infection, including blepharitis, iritis, conjunctivitis, corneal edema, and corneal infiltrates. The scoring methodology, as per Section 2.9.2, entailed grading each clinical sign on a severity scale from 0 to 3, followed by calculating a cumulative score for each eye.

#### 2.9.5. Histopathological Evaluation (HE)

At 24 h after receiving the final dose, the animals were euthanized, and their eyes were meticulously removed. The eyes were then fixed in a 10% formalin solution, embedded in paraffin, and subjected to hematoxylin and eosin (HE) staining. Using light microscopy, the resulting histological changes in the eyes were observed and analyzed.

### 2.10. Statistical Analyses

Results were expressed as the mean ± standard deviation (SD). Comparative studies of the means were carried out using a one-way analysis of variance (ANOVA). Significance was accepted with *p* < 0.05.

## 3. Results

### 3.1. OH-CATH30 Synthesis and Characterization

The OH-CATH30 peptide was synthesized manually using a solid-phase peptide synthesis technique with a high yield and purity. The purity of the peptide was confirmed by HPLC analysis, which showed a single peak with a retention time of 19.8 min (Figure 1A). The molecular weight of the OH-CATH30 peptide was determined by MALDI-TOF MS, which gave a value of 3595.80, in agreement with the calculated value of 3595.48 (Figure 1B), thus confirming the identity of the synthesized peptide. These results demonstrated the successful synthesis and characterization of the OH-CATH30 peptide.

### 3.2. Preparation of PLGA-Loaded OH-CATH30 MS

The particle size and EE% of the PLGA-OH-CATH30 MS directly affect their tissue distribution, drug content, or stability in vivo [32]. Therefore, an experiment was conducted to prepare four different types of MS by changing the osmotic pressure of the continuous phase based on an optimized formulation. The encapsulation efficiency (EE%) and particle size of the MS loaded with OH-CATH30 peptide were measured (Figure 2). The MS size exhibited a non-monotonic trend with increasing NaCl content, initially decreasing and then increasing. Similarly, the EE% also displayed a non-linear trend, initially increasing and then decreasing. The optimal conditions were obtained at 1% NaCl content, where the particle size was the smallest (459.1 ± 6.8 nm) and the EE% was the highest (74.6 ± 2.1%). Briefly, the addition of sodium chloride to the continuous phase reduces the osmotic pressure difference, resulting in a more compact internal structure in the MS that inhibits sudden drug release. Additionally, salt promotes oil droplet aggregation and PLGA deposition, leading to smaller particle sizes and higher encapsulation efficiency in the MS.

### 3.3. Characterization of OH-CATH30 MS

The optimized method was utilized to re-prepare the PLGA-OH-CATH30 MS, and their properties, including the particle size, zeta potential, polydispersity index (PDI), drug loading, encapsulation rate, and morphology, were evaluated. The size distribution of MS was assessed in an MS–aqueous solution utilizing the Zetasizer Nano ZS90. The drug delivery systems were obtained with a size range of 200 to 1000 nm and an average size of 463 ± 15.2 nm (Figure 3A), with the majority exhibiting an average diameter in the submicron range of 300 to 900 nm and a uniform particle size indicated by a PDI of less than 0.3. The zeta potential was determined to be −17.3 ± 1.91 mV, indicating a negative surface charge. This negative charge resulted in significant electrostatic repulsion between the MS, enhancing their stability in the solution and reducing the likelihood of aggregation or coalescence. The MS exhibited a smooth, spherical shape, as evidenced by scanning electron microscopy (SEM) imaging (Figure 3B). The MS achieved encapsulation efficiency of 75.2 ± 3.62% and a loading capacity of 18.25 ± 5.73% for OH-CATH30. The MS exhibited high drug loading, high yields, and a uniform size with a spherical morphology.

### 3.4. OH-CATH30 Peptide Release Profile

In our study, the release profile of PLGA-OH-CATH30 MS, as depicted in Figure 4A,B, exhibited two distinct phases of release. The first phase was an initial burst release, with a value of 45.08 ± 0.91%, occurring within the first 24 h. This burst release could be attributed to either the initial desorption of OH-CATH30 from the particle surface or the core–shell interface. The second phase of release was continuous and steady release that occurred over the following 30 days. The slower release rate during this phase may have been due to the gradual diffusion of the drug from the core to the outer layer of the MS, with the surrounding oil layer inhibiting drug release [33]. These findings suggest that the PLGA MS could potentially serve as an effective drug delivery system for the sustained release of OH-CATH-30. Finally, 75.18 ± 1.29% of the peptides encapsulated in the MS were released.

### 3.5. Bactericidal Activity

In this study, the inhibitory activity of antimicrobial peptide OH-CATH30 against *E. coli*, *S. aureus*, and *P. aeruginosa* was investigated by measuring the size of the inhibition zone on the LB medium. As shown in Figure 5A, no inhibition zone was observed when blank PLGA solution was added, while inhibition zones of significant size were observed in the presence of both antimicrobial peptide OH-CATH30 solution and PLGA-OH-CATH30 solution. This indicated that, even after PLGA embedding, the antimicrobial peptide OH-CATH30 could still cause the obvious inhibition of these strains, although the size of the inhibition zone became slightly smaller. This phenomenon can be attributed to the controlled diffusion of the antimicrobial peptide from the PLGA MS, leading to prolonged release and subsequently reducing its local concentration, which may have impacted the antimicrobial activity.

The antibacterial performance of PLGA-OH-CATH30 MS was further evaluated using the colony counting method. As shown in Figure 5B, it was observed that with the incubation time of MS and bacteria, more and more bacteria were killed, and the antibacterial efficiency gradually improved. The antibacterial efficiency of MS against Escherichia coli had already reached over 85% after incubation for 4 h, while that against Staphylococcus aureus was around 15%. However, after 24 h of incubation, the antibacterial efficiency against all three types of bacteria had reached over 90%. This may be because Staphylococcus aureus can form a protective layer called a ‘biofilm’, which makes it more difficult to be attacked by drugs and the immune system [34]. In line with previous studies [35], the antimicrobial peptides released from the MS exhibited a persistent and enhanced inhibitory effect on bacteria as the incubation time increased. This can be attributed to the gradual release of antimicrobial peptides from the MS.

### 3.6. Chorioallantoic Membrane–Trypan Blue Staining Experiment (CAM-TBS)

In a previous study [36], a significant correlation (r^2^ = 0.835; *p* < 0.0001) between the amount of absorbed dye and Draize eye irritation test scores highlighted the potential of CAM-TBS as a reliable predictor of corneal damage. In light of these findings, we employed CAM-TBS in this study to assess the cytotoxicity and biocompatibility of the prepared PLGA-OH-CATH30 MS. Figure 6A clearly demonstrates the strong irritant effect in the 0.1 M NaOH group (positive control), with significant hemorrhagic spots, hemolysis, and coagulation. In contrast, the group treated with antimicrobial peptides exhibited only slight bleeding spots, and the remaining sample groups showed no significant signs of bleeding, hemolysis, or coagulation. Furthermore, Figure 6B reveals that the uptake of trypan blue in the MS group (1.41 ± 0.19 μg) closely resembled that in the blank and saline groups. On the other hand, the OH-CATH30 group displayed a slight elevation in trypan blue uptake (3.37 ± 0.44 μg) compared to the MS group, with a statistically significant difference (*p* < 0.05). This definitive finding unequivocally confirms that the CAM in the PLGA-OH-CATH30 MS-treated group exhibited lower uptake of trypan blue compared to the directly applied peptide-treated group. This suggests that the stimulatory effect of the peptide was reduced after encapsulation, which can be attributed to the controlled and progressive release of antimicrobial peptides in drug carriers. Hence, this approach presents a gentler and safer alternative to the direct application of the peptide.

### 3.7. Ocular Tolerance Studies

In the in vivo tolerability study, all tested samples exhibited eye irritation scores below 3, as assessed by the standardized Draize eye irritation test (Table 1). Conjunctival congestion and corneal clouding scores were both zero, indicating the absence of significant irritation. Importantly, the PLGA-OH-CATH30 MS demonstrated a lower degree of irritation compared to the direct antimicrobial peptide. This difference can be attributed to the encapsulation of the antimicrobial peptide within the MS during the encapsulation process, resulting in the formation of a stable structure. This encapsulation effectively reduced direct contact and interaction between the antimicrobial peptide and ocular tissue. Moreover, the antimicrobial peptide could be released from the MS in a controlled, slow, and sustained manner, leading to a smoother and more controlled effect with reduced potential for ocular tissue irritation [37]. These findings provide further evidence supporting the safety, tolerability, and biocompatibility of our tested samples.

### 3.8. Treatment Evaluation

The study demonstrated significant and noteworthy improvements in clinical symptoms across all treatment groups, which included the OH-CATH30 solution, PLGA-OH-CATH30 MS suspension, and commercially available eye drops, when compared to the control group (Figure 7A). Specifically, the treatment group exhibited significantly reduced severity scores for conjunctival congestion, corneal edema, and infiltration, as indicated in Table 2. These findings strongly underscore the efficacy of antimicrobial peptides, regardless of whether they are administered directly or encapsulated in PLGA MS, as a highly promising therapeutic approach to combating bacterial keratitis. Moreover, the prepared MS showed comparable therapeutic efficacy to commercially available eye drops. Interestingly, the encapsulated MS displayed a slightly superior ability in reducing ocular inflammation when compared to directly administered antimicrobial peptides (AMPs), as is visually evident in Figure 7B, where the eye test scores are presented. These encouraging results highlight the potential of MS encapsulation in improving efficacy and reducing ocular inflammation.

### 3.9. Histological Analysis

The histopathological analysis was conducted on the rabbit eyes following a 5-day treatment for bacterial infection, as shown in Figure 8. In the normal group, notable pathological changes were observed in the untreated rabbits, including partial epithelial loss, extensive necrotic areas, and the formation of inflammatory granulation tissue beneath the epithelium. Within the granulation tissue, signs of neovascularization, fibroblast proliferation, and the infiltration of neutrophils, lymphocytes, and plasma cells were present (Figure 8b). Conversely, the group treated with commercially available eye drops and the PLGA-OH-CATH30 MS exhibited histological features similar to those of the normal group, indicating no significant changes in the ocular tissue (Figure 8a,c,d). In contrast, the group subjected to antimicrobial peptide treatment exhibited some modifications in the corneal stroma. Specifically, the stromal fraction displayed significant changes in the arrangement of collagen fibers, characterized by sporadic edema and the minor infiltration of lymphocytes (Figure 8e–f). Additionally, there was an observed increase in the proliferation of corneal epithelial cells and the number of cell layers. These observed alterations in the group treated with antimicrobial peptides could be attributed to the physiological response of lymphocyte infiltration during the later stages of keratitis healing, which is essential in ensuring proper immune regulation and facilitating the repair processes. It is worth noting that while antimicrobial peptides have exhibited therapeutic effects in the treatment of infections, their efficacy is significantly enhanced through encapsulation.

## 4. Discussion

The objective of this study was to utilize microsphere drug-carrier-encapsulated antimicrobial peptide technology to prepare PLGA-OH-CATH30 MS as a potential treatment for bacterial keratitis in animals. To evaluate the feasibility of these MS, we investigated their characterization, encapsulation rate, in vitro release profile, irritation potential, and in vitro antimicrobial activity. Our experimental findings demonstrated the significant antimicrobial efficacy of the PLGA-encapsulated antimicrobial peptides both in vitro and in vivo, leading to a notable reduction in infection symptoms in treated animals. Importantly, our CAM-TBS and ocular tolerance studies confirmed that the PLGA-encapsulated antimicrobial peptide MS induced significantly less irritation than administering antimicrobial peptides alone. This can be attributed to the sustained release achieved by nanomaterial encapsulation, resulting in the controlled release of the antimicrobial peptide on the ocular surface [38]. Moreover, the encapsulation of the peptides within the nanomaterials reduced direct contact between the peptides and the ocular surface, further minimizing irritation.

To enhance the embedding rate of antimicrobial-peptide-loaded MS, various modifications were made to the method described in the literature [39]. Specifically, through the optimization of various conditions, including the PLGA molecular weight, drug-to-carrier ratio, sonication time, and power, an embedding rate of approximately 50% was achieved, consistent with similar studies involving hydrophilic-peptide-loaded MS [40]. The lower embedding within the MS is mainly attributed to the higher solubility of hydrophilic drugs in the solid state, which facilitates their interaction and diffusion with polymer molecules [41]. In the context of our study, the osmotic pressure of the continuous phase was further varied to improve the embedding rate. Upon observation, the modification of the osmotic pressure not only significantly increased the embedding rate of the peptide-loaded MS but also resulted in a reduction in particle size. This notable improvement can be attributed to the effective reduction of the osmotic pressure difference between the internal and external aqueous phases by sodium chloride. Sodium chloride played a crucial role in limiting the porosity of the MS, enabling more drug encapsulation, and consequently increasing the overall encapsulation rate. Moreover, the inclusion of sodium chloride successfully reduced the swelling and fusion of droplets within the MS, ultimately leading to the formation of smaller particle sizes [42]. In the broader context, smaller particle sizes are highly favored due to their capability to provide enhanced stability and improved biodistribution profiles [43,44]. Additionally, their remarkable ability to penetrate biological barriers and selectively target specific tissues has been shown to result in improved cellular uptake, enhanced biodistribution, and ultimately increased therapeutic efficacy [45]. Moreover, specifically for corneal epithelial cells, smaller particles have been found to be associated with reduced irritation and significantly better absorption capabilities [46]. These findings underscore the paramount significance of optimizing the particle size in the successful design and implementation of medical therapeutics.

Histopathological findings revealed bacterial proliferation and corneal epithelial thickening after the administration of OH-CATH30. As reported in previous studies, OH-CATH30 selectively upregulated the production of chemokines and cytokines [47]. This stimulation of inflammatory factors promoted the wound-healing process by stimulating corneal epithelial cell proliferation [48]. A comparative analysis of pathological sections revealed that treatment with encapsulated antimicrobial peptides resulted in the restoration of the normal cell structure and thickness in corneal epithelial cells, thus preserving the corneal transparency and normal function. The sustained release mechanism of the encapsulated antimicrobial peptide therapy facilitated a rapid healing response, providing long-lasting therapeutic effects and enhancing the antimicrobial and wound-healing properties [49,50]. Our study highlights the potential of PLGA-encapsulated OH-CATH30 MS as an effective approach to promote corneal healing in bacterial keratitis. The sustained release mechanism improved the antimicrobial efficacy and reduced irritation, making these MS a valuable therapeutic option. In future research, we will intensively study the ocular pharmacokinetics of microspheres and explore their underlying mechanisms. Furthermore, we aim to further optimize the formulation, explore combination therapies, and conduct clinical trials to validate their efficacy and safety in veterinary ophthalmology.

## 5. Conclusions

In this study, we successfully prepared PLGA-loaded OH-CATH30 MS using a modified W/O/W double-emulsion method, providing significant evidence for the alleviation of bacterial keratitis infection. These MS exhibited an appropriate mean particle size, particle size distribution, and ζ-potential, demonstrating homogeneous and spherical characteristics. Additionally, they demonstrated a slow-release effect. The results of our in vitro and in vivo experiments confirmed that the PLGA MS loaded with OH-CATH30 exhibited potent antibacterial effects without causing significant irritation. Consequently, the application of OH-CATH30-PLGA MS in ocular anti-inflammatory therapy holds immense potential. Furthermore, the improved drug delivery system, achieved by altering the osmotic pressure in the continuous phase, can be extended to encapsulate other hydrophilic peptide drugs.

## Figures and Tables

**Figure 1 biomolecules-13-01244-f001:**
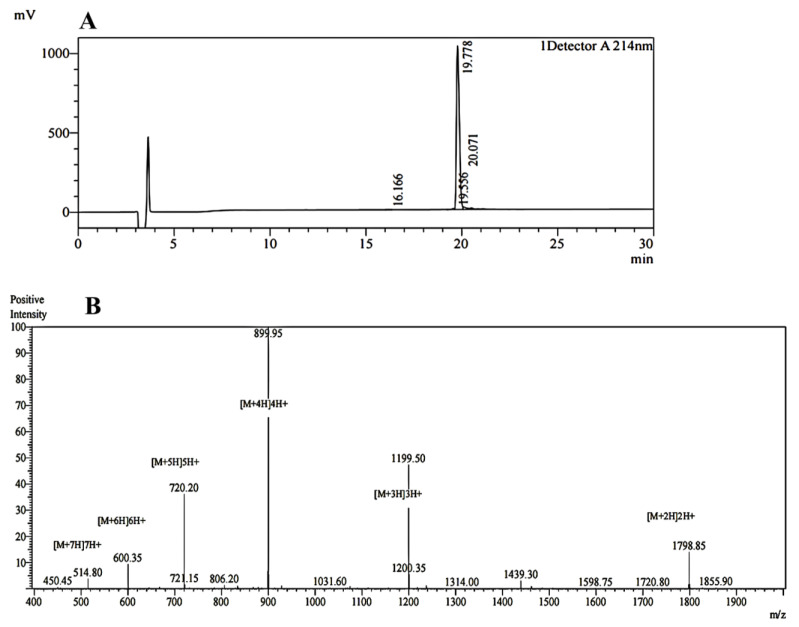
(**A**) High-performance liquid chromatography profile of OH-CATH30. (**B**) Matrix-assisted laser desorption ionization time-of-flight mass spectroscopy.

**Figure 2 biomolecules-13-01244-f002:**
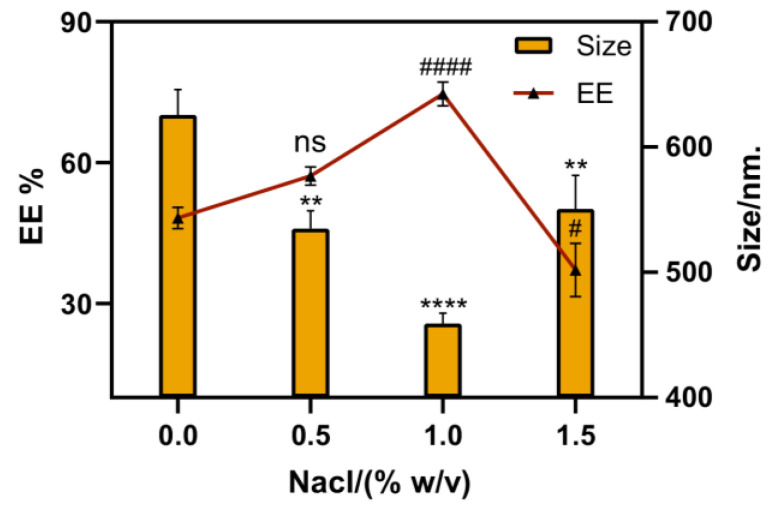
Effect of different concentrations of NaCl (% *w*/*v*) in the continuous phase on the mean particle size and EE%. Notes: Values are expressed as mean ± SD; significant differences in particle size variation were observed at different osmotic pressure conditions compared to the 0% osmotic pressure conditions (** *p* < 0.01, **** *p* < 0.0001 and "ns" indicates non-significant); significant differences in EE% variation were observed at different osmotic pressure conditions compared to the 0% osmotic pressure conditions (^#^
*p* < 0.05, ^####^
*p* < 0.0001 and "ns" indicates non-significant).

**Figure 3 biomolecules-13-01244-f003:**
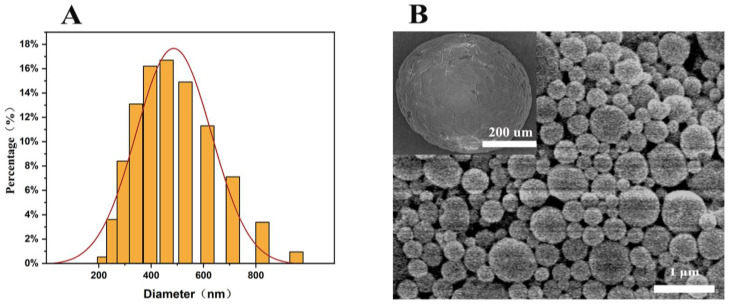
(**A**) The size distribution histogram of the MS. (**B**) Morphology of peptide-loaded PLGA MS.

**Figure 4 biomolecules-13-01244-f004:**
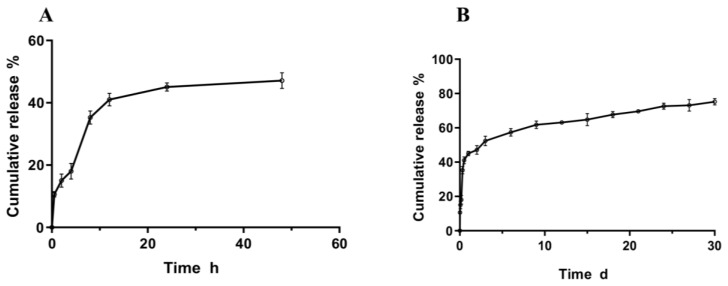
(**A**) MS release profile in PBS for 48 h. (**B**) MS release profile in PBS for 30 d. Note: Values are expressed as mean ± SD.

**Figure 5 biomolecules-13-01244-f005:**
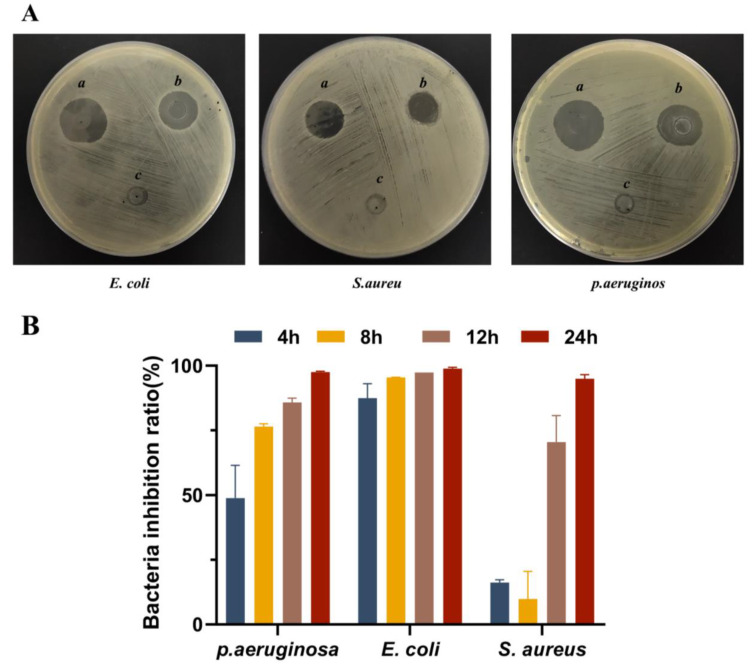
(**A**) Typical morphology of the inhibition zone induced. Notes: a, OH-CATH30; b, PLGA-OH-CATH30 MS; c, blank PLGA MS. (**B**) Antimicrobial activity of peptides released from MS against *E. coli*, *S. aureus*, and *P. aeruginosa* at 4 h, 8 h, 12 h, and 24 h. Note: Values are expressed as mean ± SD.

**Figure 6 biomolecules-13-01244-f006:**
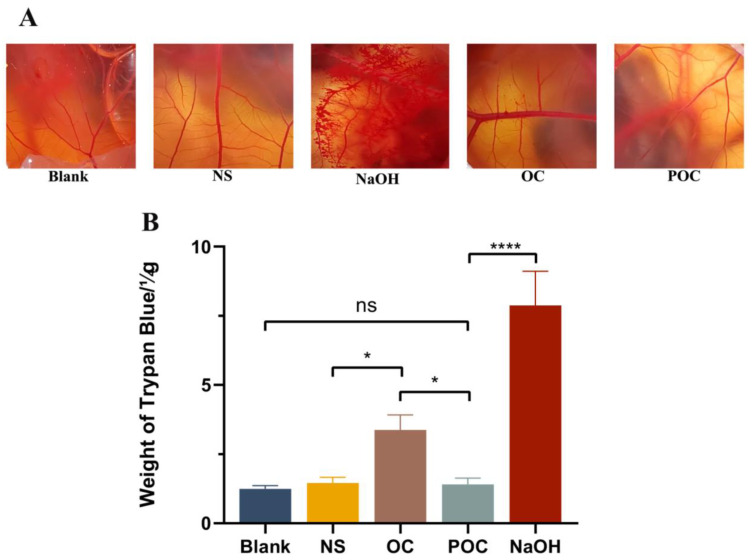
(**A**) Resulting images of CAM membranes after the PLGA-based MS administration during the HET-CAM test. (**B**) Irritation induced by antimicrobial peptides before and after encapsulation was measured by the in vitro CAM-TBS method. NaOH was used as a positive control and NS as a negative control. The difference in the trypan blue absorption rate after applying different drugs was assessed. In statistical analysis, (**** *p* < 0.0001) indicates a highly significant difference, (* *p* < 0.001) denotes a significant difference, and “ns” indicates non-significant. OC, OH-Cath30; POC, PLGA-OH-CATH30 MS.

**Figure 7 biomolecules-13-01244-f007:**
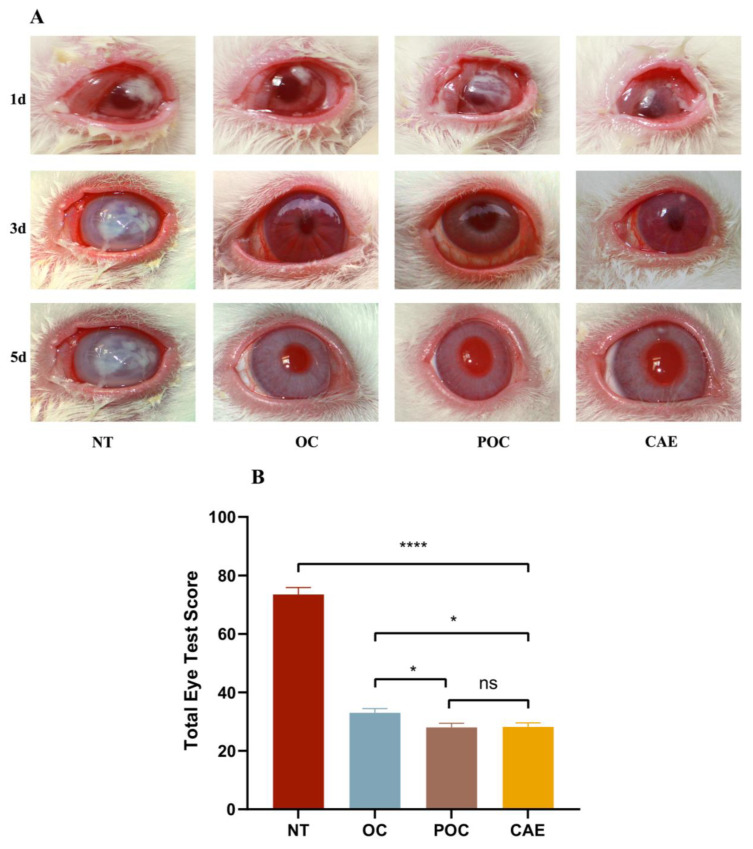
(**A**) Clinical signs of bacterial keratitis in the control group, the groups before and after the encapsulation of antimicrobial peptides, and the group treated with commercially available eye drops. (**B**) Clinical rabbit keratitis total eye test scores after treatment with different medications. The changes in the total scores of rabbit eyes treated with different drugs were extremely significant (**** *p* < 0.0001) compared to the untreated group. No significant(ns) difference was observed between commercially available eye drops and PLGA-OH-CATH30, while a significant (* *p* < 0.05) difference was found between OH-Cath30 and commercially available eye drops. NT, no treatment; OC, OH-Cath30; POC, PLGA-OH-CATH30; CAE, commercially available eye drops.

**Figure 8 biomolecules-13-01244-f008:**
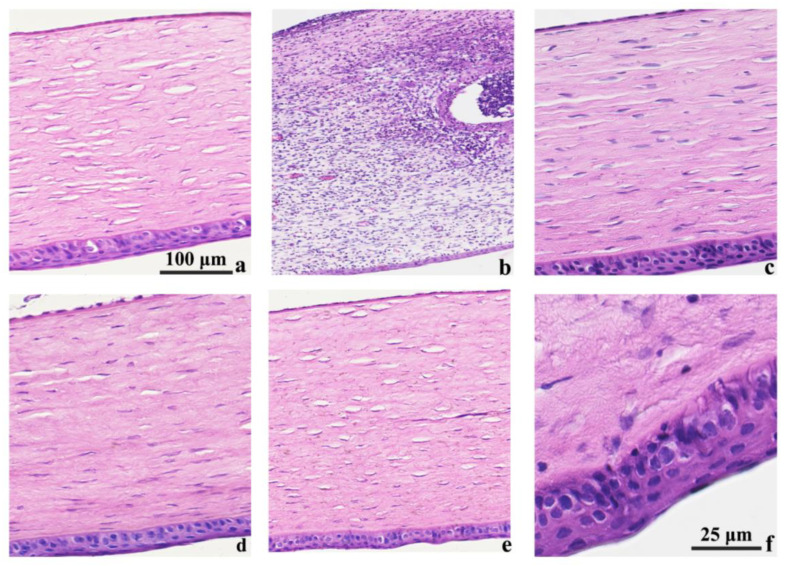
Histological examination of the corneas of rabbit eyes treated for 5 days (magnification: 200×). (**a**) Uninfected; (**b**) untreated after infection; (**c**) commercially available pet-specific Tarivid ophthalmic solution (containing 1 mg/mL of Ofloxacin) treatment; (**d**) PLGA-OH-CATH30 nanoparticle treatment; (**e**) OH-CATH30 peptide treatment; (**f**) OH-CATH30 peptide treatment (enlarged). Figures (**a**–**e**) share a common scale.

**Table 1 biomolecules-13-01244-t001:** Macroscopic sign grading system in nanoparticle tolerance studies in vivo.

Formulations	Adverse Effect	Test Score							
		Time (h)							
		0.5	1	2	4	12	24	48	72
Saline	Discharge	0	1	0	0	0	0	0	0
	Cornea	0	0	0	0	0	0	0	0
	Conjunctiva	0	0	0	0	0	0	0	0
	Lids	0	0	0	0	0	0	0	0
Commercially available eye drops	Discharge	0	0	1	0	0	0	0	0
	Cornea	0	0	0	0	0	0	0	0
	Conjunctiva	0	0	0	0	0	0	0	0
	Lids	0	0	1	0	0	0	0	0
OH-CATH30	Discharge	0	0	1	0	0	0	0	0
	Cornea	0	0	0	0	0	0	0	0
	Conjunctiva	0	0	0	0	0	0	0	0
	Lids	0	1	1	0	0	0	0	0
PLGA-OH-CATH30	Discharge	0	0	0	1	0	0	0	0
	Cornea	0	0	0	0	0	0	0	0
	Conjunctiva	0	0	0	0	0	0	0	0
	Lids	0	0	0	0	0	0	0	0

**Table 2 biomolecules-13-01244-t002:** In vivo antibacterial activity scores of antimicrobial peptides before and after encapsulation.

Days		1	2	3	4	5
Redness	NT	3.96 ± 0.23	3.37 ± 0.21	3.47 ± 0.45	3.45 ± 0.25	3.77 ± 0.25
	OC	3.57 ± 0.40	2.27 ± 0.45	1.36 ± 0.2	0.83 ± 0.25	0.37 ± 0.03
	POC	3.43 ± 0.38	2.03 ± 0.25	1.03 ± 0.15	0.30 ± 0.44	0.06 ± 0.02
	CAE	3.47 ± 0.38	2.09 ± 0.17	1.01 ± 0.26	0.22 ± 0.30	0.05 ± 0.01
Lacrimal secretion	NT	1.86 ± 0.1	1.93 ± 0.12	1.87 ± 0.32	1.83 ± 0.06	1.83 ± 0.06
	OC	1.73 ± 0.32	1.13 ± 0.12	0.97 ± 0.06	0.29 ± 0.05	0.13 ± 0.02
	POC	1.5 ± 0.17	0.88 ± 0.26	0.62 ± 0.26	0.06 ± 0.01	0.00 ± 0.00
	CAE	1.67 ± 0.42	0.84 ± 0.3	0.61 ± 0.36	0.07 ± 0.05	0.00 ± 0.00
Mucoid discharge	NT	2.93 ± 0.32	2.83 ± 0.31	2.73 ± 0.21	2.77 ± 0.23	2.83 ± 0.10
	OC	2.37± 0.2	1.55 ± 0.26	0.97 ± 0.32	0.67 ± 0.07	0.00 ± 0.00
	POC	2.17 ± 0.31	1.34 ± 0.15	0.62 ± 0.12	0.06 ± 0.03	0.00 ± 0.00
	CAE	2.21 ± 0.44	1.67 ± 0.36	0.51 ± 0.1	0.07 ± 0.02	0.00 ± 0.00
Response to ocular stimulus	NT	3.87 ± 0.72	3.53 ± 0.32	2.27 ± 0.25	1.9 ± 0.1	1.81± 0.26
	OC	3.53 ± 0.12	2.36 ± 0.26	0.83 ± 0.35	0.22 ± 0.06	0.00 ± 0.00
	POC	3.32 ± 0.06	2.15 ± 0.17	0.68 ± 0.31	0.07 ± 0.05	0.00 ± 0.00
	CAE	3.27 ± 0.06	2.07 ± 0.15	0.53 ± 0.15	0.09 ± 0.02	0.00 ± 0.00
Swelling of the eyelid	NT	3.72 ± 0.32	3.97 ± 0.06	3.64 ± 0.15	3.64 ± 0.06	3.71 ± 0.26
	OC	3.27 ± 0.06	2.77 ± 0.36	1.77 ± 0.31	0.07 ± 0.31	0.00 ± 0.00
	POC	3.7 ± 0.26	2.73 ± 0.2	1.43 ±0.03	0.03 ± 0.01	0.00 ± 0.00
	CAE	3.57 ± 0.35	2.82± 0.2	1.35 ± 0.4	0.00 ± 0.00	0.00 ± 0.00

Values are reported as means ± SD (*n* = 6). NT, no treatment; OC, OH-Cath30; POC, PLGA-OH-CATH30; CAE, commercially available eye drops.

## Data Availability

Not applicable.

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
