# Peer review of "Assessing the Efficacy of PLGA-Loaded Antimicrobial Peptide OH-CATH30 Microspheres for the Treatment of Bacterial Keratitis: A Promising Approach"

_biomolecules, 2023, doi:10.3390/biom13081244_

Round 1

Reviewer 1 Report

The manuscript presents an interesting approach to managing keratitis in animals. It is interesting to note the efficacy of the peptide chosen and the use of NaCl to improve encapsulation efficiency of hydrophilic peptide. 

Methods and results section can be improved to provide better clarity to the reader. Specific comments are:

Please check in-text citations. Some incorrect formatting e.g. J.E. Rosas et al should be Rosas et al. ; Young Hwan Choi et al should be Choi et al.

Line 48 – “nanotechnology has proposed an innovative approach for the controlled release and protection of antimicrobial peptides” do you mean nanotechnology provides an innovative approach.

Section 2.2 – should provide further details on analysis methods e.g. HPLC mobile phase, column details, detection mode etc

Line 100 – W1/O/W2 can be written as W/O/W

Line 147 – plga should be PLGA

Line 169 – remove the words “by reference” and revise to state “as previously described”

Section 2.9.2 – should include concentration of the OH-CATH30 investigated

Section 2.9.3 – what is the commercially available eye drop formulation used? Please specify. How long after the corneal infection induction did the treatment start? Was this immediately after injecting s.aureus into corneal stroma? How did you confirm keratits or corneal infection and ensure each rabbit had similar baseline scores? Please clarify the reasons for using NP formulation three times a day if this provides sustained release of the peptide.

Figure 2 – what is the unit for size ? Please check axis title e.g. nacl should be NaCl (% w/v)

Section 3.3 – methods should provide how loading efficiency is calculated.

Section 3.4 – authors mention that 75% of encapsulated peptide is released at 30 days. Can you confirm the stability of peptide during this time. Ideally should conduct LC-MS to confirm the stability at 30 days at 37C.

Table 1 – what is the first row formulation? Are the results provided for control? If so please specify this in the table.

Figure 8 – should include scale bars. What is 8C – is this is the commercial preparation? Also what is the different between 8e and 8f? both appear to be just the peptide formulation.

Table 2 – are there any statistically significant differences between OC and POC? How can the authors confirm that the encapsulation enhance therapeutic effects of the peptide? (line 408) – this is not evident from in-vivo data.

Discussion – authors report that NP formulation is significantly less irritant than the peptide alone. It is unclear how authors derive this from data presented.

please proof read manuscript to improve clarity and flow of content. 

Reviewer 2 Report

The manuscript biomolecules-2526148 ''Assessing the Efficacy of PLGA-Loaded Antimicrobial Peptide OH-CATH30 Microspheres for the Treatment of Bacterial Keratitis: A Promising Approach'' by Xiao-Qian Jiao et al. describes the preparation of a biocompatible and biodegradable PLGA-based nanocarrier for encapsulating the antimicrobial peptide OH-CATH30 for the treatment of bacterial keratitis in animals. The obtained particles were comprehensively characterized both in vitro (including encapsulation efficiency, release kinetics) and in vivo/vitro (antimicrobial activity and toxicity). The developed systems had improved bioavailability and high antibacterial efficacy; they are promising therapeutic agents for topical application in the treatment of keratitis instead of conventional antibiotics. The topic of the study is certainly relevant. The manuscript is logical and well written. The paper will definitely be of interest to the readers of Pharmaceutics. Nevertheless, there are crucial questions for discussion.

  1. You call the obtained particles microspheres in the title of the paper, but nanoparticles in the text of the paper. It seems to me that it is better to call them as microspheres or polymer particles, since for nanoparticles they are large in size. 

  2. The drug release condition is 370C, although the ocular surface temperature is about 320C, and this may affect the release rate.

  3. What method did you use to calculate the polydispersity index?

  4. The agar gel diffusion method was used to test the antimicrobial activity of in vitro. However, I assume that the peptide will adhere to the agar, thus distorting the results.

  5. Correct the CFU/mL values - Line 140 and Line 152.

  6. Why is a 30-day prolonged release required for an ocular dosage form for topical application? Obviously, the nanocarrier will be removed from the corneal surface more quickly by tear fluid. Anyway, I suggest examining the mucoadhesive properties of the obtained polymer particles.

Minor editing of English language required

Reviewer 3 Report

The study of Xiao-Qian Jiao, Xu-Feng Dong, Hu Shan, and Zhi-Hua Qin " Assessing the Efficacy of PLGA-Loaded Antimicrobial Peptide 2 OH-CATH30 Microspheres for the Treatment of Bacterial Kera-3 titis: A Promising Approach" was aimed at the development of poly (lactic-co-9 glycolic acid) encapsulated nanoparticles of the naturally occurring antimicrobial peptide OH-CATH30.

In this study the advantage of the synthesized has been proved in vitro and in vivo for the alleviation of bacterial keratitis infection.

The findings of the study are meaningful and can be considered as a new step in strengthening potent antibacterial effects without causing significant irritation.

The manuscript is clear, well-structured, the methods are thoroughly described. The in vitro and in vivo findings are discussed together. The reference list contains a considerable amount of recent investigations.

In a whole, the manuscript makes good impression and can be published in "Biomolecules" Journal after addressing the following issues:

Comments and recommendations:

Lines 72-74

In this sentence it is unclear "these membranes" - rewrite or explain.

Line 121

That was the accuracy of determination of the particle size and mean diameter distribution (polydispersity index)? Add, please.

Line 122-134

NPs in vitro release experiment

Once again, the accuracy of in vitro release experiment should be indicated.

Line 273

Besides the surface charge the zeta potential serves as an indicator of the system stability.

Some additional discussion on this issue is recommended.

Line 293

Add A and B to the plots in Figure 4.

Line 438

The importance of smaller particle size for medical use should be emphasized and supported by the appropriate referemnce.

Misprints:

Line 65

Remove a comma after [22].

Lines 63, 65, 68…79…

Correct the intervals between the text and cited references throughout the manuscript.

Ex.: "…….treatment [22] or "….drugs either encapsulated in the matrix or adsorbed on the surface[20,21]."

Line 271

Correct the interval

Round 2

Reviewer 2 Report

The manuscript may be accepted.

English is fine

Author Response

Thanks for your review.